# Comparative Efficacy of Tapentadol versus Tapentadol Plus Duloxetine in Patients with Chemotherapy-Induced Peripheral Neuropathy (CIPN): A Randomized Non-Inferiority Clinical Trial

**DOI:** 10.3390/cancers14164002

**Published:** 2022-08-18

**Authors:** Pasquale Sansone, Luca Gregorio Giaccari, Caterina Aurilio, Francesco Coppolino, Maria Beatrice Passavanti, Vincenzo Pota, Maria Caterina Pace

**Affiliations:** Department of Woman, Child and General and Specialized Surgery, University of Campania “Luigi Vanvitelli”, 80138 Naples, Italy

**Keywords:** chemotherapy-induced peripheral neuropathy, duloxetine, tapentadol

## Abstract

**Simple Summary:**

Chemotherapy-induced peripheral neuropathy (CIPN) is a common complication due to treatment with anti-cancer agents. Our aim was therefore to assess the non-inferiority of the analgesic effect and safety of tapentadol alone compared to duloxetine plus tapentadol administered to patients with chemotherapy-induced peripheral neuropathy. The use of tapentadol is a safe and effective analgesic therapy in patients with CIPN. Positive effects of tapentadol were noted on the patients’ quality-of-life assessments.

**Abstract:**

Introduction: Chemotherapy-induced peripheral neuropathy (CIPN) is a common complication due to treatment with many commonly used anti-cancer agents. CIPN is a mainly sensory neuropathy that can be characterized by the appearance of motor and autonomic alterations. Clinicians may offer duloxetine (DLX) for patients with cancer experiencing CIPN. Our aim was to assess the non-inferiority of the analgesic effect and safety of tapentadol (TP) alone compared to duloxetine plus tapentadol administered to patients with CIPN. Methods: A total of 114 patients were enrolled in the study and randomized to receive tapentadol in a dosage of 50 to 500 mg/day (n = 56) or tapentadol plus duloxetine in a dosage of 60 to 120 mg/day (n = 58) for a period of 4 weeks. We evaluated the analgesia efficacy, defined as a decrease in pain on the NRS between the first administration and 28 days later. Secondary endpoints included analgesia efficacy at 28 and 42 days, defined by a decrease in DN4 and LEPs, decrease in quality of life, and the incidence of any serious or non-serious adverse events after the first administration. Results: In this randomized, double-blind trial comparing TP and TP plus DLX for CIPN management, TP was feasible and non-inferior to the association with DLX as far as the reduction of pain after chemotherapy at 28 days is concerned. Scores on other rating scales evaluating the quality of life, anxiety and depression, and the characteristics of pain revealed similar improvements associated with tapentadol versus duloxetine at these time points. Conclusion: The use of TP is a safe and effective analgesic therapy in patients with CIPN. Positive effects of TP were noted on the patients’ quality-of-life assessments.

## 1. Introduction

### 1.1. Background

Chemotherapy-induced peripheral neuropathy (CIPN) is a common complication due to treatment with many commonly used anti-cancer agents. The antineoplastic agents associated with CIPN are platinum-based drugs (e.g., carboplatin, cisplatin and oxaliplatin), taxanes (e.g., paclitaxel and docetaxel), epothilones (e.g., ixabepilone), vinca alkaloids (e.g., vincristine and vinblastine), bortezomib, and thalidomide [1,2].

The high success rate of these drugs in cancer treatment has led to a steady increase in the survival rates of patients. As a result, the number of cancer survivors suffering from neuropathic pain conditions is also increasing. CIPN prevalence is 68.1% in the first month after chemotherapy, 60.0% at 3 months, and 30.0% at 6 months or later [3].

CIPN is a mainly sensory neuropathy that can be characterized by the appearance of motor and autonomic alterations [4]. Sensory symptoms are usually the first to appear, involving the feet and hands and commonly presenting themselves as a typical “glove and stocking” neuropathy [5]. The symptoms include pain, numbness, and tingling [6]. Motor symptoms are usually weakness, gait, balance disturbances, and impaired movements, but they occur less frequently than sensory symptoms [6]. Autonomic disorders involve orthostatic hypotension, constipation, and altered sexual or urinary function [6].

Risk factors for developing CIPN are diabetes mellitus, alcohol abuse, or inherited neuropathy. Thyroid dysfunction; HIV infection; vitamin B1, B6, and B12 deficiencies; and monoclonal gammopathy have also been reported in the pathogenesis of CIPN [7].

According to the American Society of Clinical Oncology (ASCO) guidelines, clinicians may offer duloxetine (DLX) for patients with cancer experiencing CIPN [8]. It inhibits the reuptake of serotonin and norepinephrine in the central nervous system. ASCO revised different CIPN trials concerning the use of various treatments, such as tricyclic antidepressants (such as nortriptyline), gabapentin, amitriptyline, and ketamine. Trials are inconclusive regarding these treatments, but according to the authors, these agents may be offered on the basis of data supporting their utility in other neuropathic pain conditions given the limited number of other CIPN treatment options.

Tapentadol (TP) is a centrally acting opioid analgesic with a dual mode of action as an agonist of the μ-opioid receptor (MOR) and as a norepinephrine reuptake inhibitor (NRI) [9,10]. TP was developed for the management of moderate–severe chronic pain. It may be beneficial in the treatment of painful peripheral neuropathies (PPN), such as diabetic painful neuropathy and postherpetic neuralgia [11]. A recent study has indicated TP as a candidate in the management of patients suffering from the neuropathic pain of CIPN also as a first line of therapy [12]. Thirty-one patients with moderate-to-severe neuropathic pain from CIPN were treated with TP. It was well-tolerated and effective in the treatment of neuropathic pain.

### 1.2. Aims

Our aim was therefore to assess the non-inferiority of the analgesic effect and safety of tapentadol alone compared to duloxetine plus tapentadol administered to patients with chemotherapy-induced peripheral neuropathy.

The primary endpoint of this trial was to verify the non-inferiority of TP in terms of efficacy, considering pain intensity after 28 and 42 days of treatment. As a secondary endpoint, we aimed to examine the effects on the immediate quality of recovery in relation to adverse effects and patient satisfaction.

## 2. Patients and Methods

### 2.1. Study Design

This was a single-center, randomized, double-blind, controlled, non-inferiority trial. The full protocol for this trial is available upon request. We completed this trial between May 2018 and November 2019 at the “Pain Unit” of the University of Campania “Luigi Vanvitelli” (I° Policlinico, Hall 3 WEST, 2nd floor—Piazza Miraglia, 5—80138 Naples, Italy). This study was approved by the local institutional review board (Ethics Committee Code 1154/12) and conducted according to the Declaration of Helsinki. All patients provided written informed consent. This study is reported as per the Consolidated Standards of Reporting Trials (CONSORT) guideline [13].

### 2.2. Patient Population

Patients reporting pain related to chemotherapy, based on medical history and clinical examination, and lasting more than 1 month were screened in the study. We considered only patients aged more than 18 years able to provide written informed consent.

Patients were excluded if they met any of the following criteria: neuropathy due to other diseases; previous treatment of pain in the week before starting the study; other non-medical intervention/therapy for the treatment of peripheral neuropathy; liver, kidney, or heart failure; contraindications to the drugs to be tested (e.g., TP and/or DLX allergy); and women who were breastfeeding or pregnant (excluded from medical history and/or laboratory tests).

### 2.3. Randomization and Masking

Patients were randomized via computer-generated codes to receive TP or TP plus DLX. All medical staff, as well as patients, were blinded to the treatment group, with the exception of the pharmacist responsible for preparing the blisters. Treatments were prepared in the hospital pharmacy and the investigator was given blisters labelled only “A” and “B”. The randomization (1:1) in parallel groups was pre-performed in blocks of random size using a dedicated software available to the University of Campania “Luigi Vanvitelli”. The randomization list was delivered to the central pharmacy in charge of preparing the packs.

### 2.4. Procedures

In line with normal prescribing practice, patients received tapentadol in a dosage of 50 to 500 mg/day for a period of 4 weeks (***TP group***) or tapentadol plus duloxetine in a dosage of 60 to 120 mg/day for a period of 4 weeks (***DLX group***).

In the TP group, an initial dose of 50 mg of tapentadol was administered. An additional dose of 50 mg was administered at 1 h if the Numerical Rating Scale (NRS) remained > 3/10.

In the DLX group, an initial dose of 60 mg of duloxetine and 50 mg of tapentadol were administered.

In both groups, pain relief was assessed every 7 days to determine whether titration of the drug was necessary. Patients with NRS < 3/10 continued to receive the starting dose. In all other cases, the dose was increased gradually.

If an intolerable adverse event (AE) occurred, symptomatic treatment was guaranteed, and the dose of the drug was eventually reduced.

Patients could not switch between groups until the end of the study; however, they may have chosen to leave the studio for any reason at any time.

### 2.5. Data Collections and Outcomes

Data collections and outcomes were completed by research members who were trained before the study and not involved in the management of the patients.

The primary endpoint was analgesia efficacy, defined as a decrease in pain on the NRS between the first administration and 28 days later. Pain intensity was evaluated also at 42 days.

Secondary endpoints included analgesia efficacy at 28 and 42 days, defined by a decrease in DN4 and LEPs, decrease in quality of life, and the incidence of any serious or non-serious adverse events after the first administration.

Pain intensity was evaluated using the 11-point **Numerical Rating Scale** (**NRS**) from 0 (no pain) to 10 (worst possible pain), based on average pain over the past 24 h [14]. The **Douleur Neuropathique 4** (**DN4**) was used for the diagnosis of neuropathic pain if the total score was greater than or equal to 4 [15].

Several techniques are available for the objective evaluation of pain in patients. The **laser-evoked potentials** (**LEPs**) is a useful tool to assess conduction via nociceptive nerve fibers Aδ and C and can be used to identify nerve-fiber dysfunction and altered nociception [16,17,18,19]. A MYOQUICK matrix line (Micromed SpA, Treviso, Italy), a system containing a stimulatory neodymium-doped yttrium aluminum garnet (Nd:YAG) laser of wavelength 1064 nm, was used to produce LEPs. The laser pulses were applied at the site where neuropathic pain was reported. Nerve impulses were generated with a series of 40 stimulations in 10 min with an interval between stimulations of 10–20 s. At 3 s after each stimulation, the patients expressed pain level using NRS; for comparison, a value of 4 was assigned to the pain sensation after a pinprick. An electroencephalogram was recorded using a BRAIN QUICK system (Micromed SpA). Electrodes were applied to the scalp in the frontal, central, and parietal midline positions, according to the international 10–20 positioning system. The latencies and amplitudes of the N2 and P2 components of the LEPs were recorded at the central midline (Cz) position, where the LEP signal is maximum.

The severity of cancer-related pain is affected by the development of catastrophic thoughts in patients, such as “my pain will never improve without a doubt”. This effect was evaluated using the **Pain Catastrophizing Scale** (**PCS**) [20].

The effects of pain on patients’ psychiatric symptoms (anxiety and depression) were assessed using the **Hospital Anxiety and Depression Scale** (**HADS**) [21]. A HADS score of <20 points was used as a limit for detecting patients with severe depression.

General health status was assessed using the European Organization for the Research and Treatment of Cancer Quality of Life Chemotherapy-Induced Peripheral Neuropathy Questionnaire (EORTC QLQ-CIPN20) [22].

Adverse events (AEs) were monitored throughout the study, particularly drowsiness, dizziness, nausea, palpitations, and hypertension. **Common Terminology Criteria for Adverse Events** (**CTCAE**)—Version 5.0. was utilized for AEs reporting [23].

Finally, a self-assessment was performed by patients using **Pain Relief Scale** (**PRS**) [24]. Patients determined the effectiveness of TP or TP plus DLX themselves using a 4-point scale: “complete relief”, “much relief”, “mild relief”, and “no change”.

During the first visit (**randomization time, T0**) the following data were carried out:Remote and proximal anamnestic collection.Physical and documentary examination.Quantitative Rating Scale (NRS) and Qualitative Pain Assessment (Questionnaire DN4).Hospital Anxiety and Depression Scale (HADS).Assessment of quality of life (Quality of Life, QoL) through the European Organization for the Research and Treatment of Cancer Quality of Life Chemotherapy-Induced Peripheral Neuropathy Questionnaire (EORTC QLQ-CIPN20).

The essential clinical features of the CIPN to be researched included:Distal involvement of peripheral nerves with typical “glove and stocking distribution”.Symmetric distribution.Temporal onset after administration of chemotherapy drugs.Signs and symptoms of sensorineural dysfunction (i.e., paraesthesia, dysaesthesia, hyperaesthesia or hypoaesthesia) and pain.Relative savings of motor function with mild to moderate motor weakness, which may be accompanied by distal distribution myopathy.Severity of sensory symptoms reported disproportionate in relation to motor symptoms.

The following evaluations were performed in 4 time points:7 days (day 7, **T7**) after the start of treatment.14 days (day 14, **T14**) after the start of treatment.21 days (day 21, **T21**) after the start of treatment.28 days (day 28, **T28**) after the start of treatment.

The evaluation included:Physical and documentary examination.Quantitative Rating Scale (NRS) and Qualitative Pain Assessment (Questionnaire DN4).Pain Relief Scale (PRS).Laser Evoked Potentials (LEPs).Pain Catastrophizing Scale (PCS).Hospital Anxiety and Depression Scale (HADS).Quality of Life (QoL) assessment through the European Organization for the Research and Treatment of Cancer Quality of Life Chemotherapy-Induced Peripheral Neuropathy Questionnaire (EORTC QLQ-CIPN20).Adverse Events (AEs).

The follow-up phase included the evaluation of the patients at 42 days (follow-up time, **T42**) after the start of treatment.

### 2.6. Sample Size and Statistical Analysis

Our study was designed to demonstrate the non-inferiority of TP to TP plus DLX as far as the change in average pain intensity from baseline to the 28th day of the double-blind treatment period is concerned. As a non-inferiority study, the threshold was based on a clinically meaningful difference in efficacy (NRS_T28_–NRS_T0_) between the two treatments of −1.5. From previous studies, the standard deviation of the reduction in pain was estimated to be 2.5.

With a sample size of 90 patients (45 in each group), a 2-group design would have provided at least 80% power to reject the null hypothesis (corresponding to a loss of efficacy greater than or equal to 1.5), in favor of the alternative hypothesis, corresponding to a gain in efficacy or a loss of efficacy of less than 1.5, assuming that the expected mean difference was 0, the common standard deviation 2.5, and the level of significance 2.5%. Assuming a drop-out rate of 10%, a total sample size of 100 patients was required.

Non-inferiority was determined on the basis of a 1-sided mean-equivalence *t*-test (two 1-sided t-tests approach, TOSTT procedure) [25] on the per-protocol population (primary endpoint: NRS efficacy [NRS_T28_–NRS_T0_]). We also compared the efficacy of the TP treatment to TP plus DLX treatment at the different time points (NRS_T7_–NRS_T0_, NRS_T14_–NRS_T0_, NRS_T21_–NRS_T0_ and NRS_T42_–NRS_T0_) in the per-protocol population using a two-sample Student’s *t*-test.

An analysis of variance (ANOVA) with repeated measures was performed for DN4, HADS, QLQ-CIPN20, PCS, and N2 and P2 latency and amplitude. A chi-squared test was performed for the PRS score.

We compared the incidence of serious and non-serious adverse events between the groups using a chi-squared test.

Demographic and baseline characteristics were compared by a two-sample Student’s *t* test for numeric variables.

All statistical analyses were performed using SAS^®^ version 9.3 (SAS Institute srl, Milan, Italy).

## 3. Results

### 3.1. Patient Characteristics

A total of 114 patients were enrolled in the study and randomized to receive tapentadol in a dosage of 50 to 500 mg/day (n = 56) or tapentadol plus duloxetine in a dosage of 60 to 120 mg/day (n = 58) for a period of 4 weeks (Figure 1). Doses were titrated to adequate pain relief or dose-limiting toxicity on the basis of the clinical response. A total of 108 patients completed the trial. A total of 92.8% (52/56) of patients in the TP group and 96.5% (56/58) of patients in the TP plus DLX group completed the double-blind treatment period; the most common reasons for treatment discontinuation in both treatment groups were a previous treatment (TP group, 1.9% [1/52]; TP plus DLX group, 1.8% [1/56]) and consent withdrawal (TP group, 1.9% [1/52]; TP plus DLX group, 1.8% [1/56]). As a result, 52 patients in the TP group and 56 in the TP plus DLX group constituted the protocol population.

Demographic and baseline characteristics were comparable between the treatment groups in the overall population, as shown in Table 1. At baseline, no major differences were noted between the two study groups regarding age (52.4 vs. 51.7 years) and sex (20/32 vs. 22/34 male/female).

Patient tumour characteristics of the two groups were well balanced. In the TP group, forty-seven patients (90.4%) had a solid tumour, and the most frequent were breast (40.4%) and digestive system cancer (27.6%). Forty-six patients (88.5%) underwent surgery, and twenty-one patients (40.4%) received radiotherapy. The majority of patients were affected by taxane-induced CIPN (42.3%), followed by platinum-compound-induced CIPN (34.6%). In the TP plus DLX group, eighteen patients (36.7%) presented breast cancer and fourteen (28.6%) digestive cancer. Globally, forty-nine patients (87.5%) had a solid tumour. Surgical treatment was performed forty-nine times (87.5%), while radiotherapy twenty-two times (39.3%). Similarly to the other group, CIPN was diagnosed in twenty-four patients (42.8%) treated with taxanes and in nineteen patients (33.9%) treated with platinum-based drugs.

The duration of treatment was 28 days in both the TP and TP plus DLX treatment groups. The median of the mean total daily dose (TDD) of study drug taken during the treatment was 104.5 mg for tapentadol in the TP group and 51.6 mg for tapentadol plus 78.6 mg for duloxetine in the TP plus DLX group.

### 3.2. Primary Endpoint

#### Pain Intensity

At baseline, all patients showed a NRS ≥ 4. We considered NRS higher than 4 as the optimal cut-off point between mild and moderate pain [26]. This cut-off was identified as the threshold of tolerable pain.

The mean difference between NRS at first administration and NRS at 28 days was −4.21 in the TP group and −4.4 in the TP plus DLX group, with a mean difference in the NRS variation between the groups of 0.19.

At days 7, 14, 21, 28, and 42, the mean NRS score was significantly (*p* < 0.05) reduced from baseline in both the TP and TP plus DLX groups. The reduction in the NRS score from baseline to day 7 (−0.5 vs. −0.63; *p* = 0.96), day 14 (−1.41 vs. −1.6; *p* = 0.92), day 21 (−2.88 vs. −3.17; *p* = 0.88), day 28 (−4.21 vs. −4.4; *p* = 0.90), and day 42 (−4.46 vs. −4.64; *p* = 0.93) was not significantly different in the TP group compared to the TP plus DLX group (see Table 2 and Figure 2).

### 3.3. Secondary Endpoints

#### 3.3.1. Douleur Neuropathique 4

At the initial evaluation, all patients showed a DN4 ≥ 4, suggesting neuropathic pain [27]. At days 7, 14, 21, 28, and 42, the mean DN4 score was significantly (*p* < 0.05) reduced from baseline in both the TP and TP plus DLX groups. The reduction in the NRS score from baseline to day 7 (−0.81 vs. −0.96; *p* = 1.01), day 14 (−2.34 vs. −2.86; *p* = 0.76), day 21 (−3.71 vs. −3.83; *p* = 0.98), day 28 (−4.6 vs. −4.76; *p* = 1.03), and day 42 (−5.03 vs. −5.21; *p* = 1.02) was not significantly different in the TP group compared to the TP plus DLX group (Table 2 and Figure 3).

#### 3.3.2. Laser Evokated Potentials

At day 28, mean N2 latency and amplitude and mean P2 latency and amplitude were all significantly (*p* < 0.05) lower from baseline in both the TP group and TP plus DLX groups (Table 2). Mean changes from baseline to day 28 were similar in both groups for the following parameters: N2 latency (−32.8 vs. −33.5 ms; *p* = 1.14); N2 amplitude (+13.5 vs. +12.7 µV; *p* = 1.02); P2 latency (−30.5 vs. −32.8 ms; *p* = 1.24); and P2 amplitude (+11.8 vs. +11.3 μV; *p* = 1.06).

#### 3.3.3. Hospital Anxiety and Depression Scale

At days 7, 14, 21, 28, and 42, the mean HADS score was significantly (*p* < 0.05) reduced from baseline in both the TP and TP plus DLX groups. The reduction in the HADS score from baseline to day 7 (−2.11 vs. −2.15; *p* = 1.26), day 14 (−4.28 vs. −4.54; *p* = 0.91), day 21 (−6.11 vs. −5.99; *p* = 1.01), day 28 (−8.6 vs. −8.3; *p* =0.88), and day 42 (−9.14 vs. −8.69; *p* = 0.81) was not significantly different in the TP group compared with the TP plus DLX group (Table 2).

#### 3.3.4. Quality of Life

At days 7, 14, 21, 28, and 42, the mean QLQ-CIPN20 score was significantly (*p* < 0.05) reduced from baseline in both the TP and TP plus DLX groups. The reduction in the QLQ-CIPN20 score from baseline to day 7 (−6.18 vs. −7.92; *p* = 0.92), day 14 (−17.37 vs. −20.01; *p* = 0.85), day 21 (−24.88 vs. −28.94; *p* = 0.82), day 28 (−36.31 vs. −38.24; *p* = 0.90), and day 42 (−40.7 vs. −43.55; *p* = 0.86) was not significantly different in the TP group compared to the TP plus DLX group.

#### 3.3.5. Pain Relief Scale

At day 28, tapentadol and duloxetine recipients reported “complete relief” (51.9 vs. 53.6%), “much relief” (38.5 vs. 32.1 %), or “mild relief” (9.6 vs. 14.3%). No patient reported “no change” after the treatment. At day 42, tapentadol and duloxetine recipients reported “complete relief” (67.3 vs. 69.6%), “much relief” (30.8 vs. 28.6%), or “mild relief” (1.9 vs. 1.8%). No patient reported “no change” after the treatment (Table 2).

#### 3.3.6. Pain-Catastrophizing Scale

At days 7, 14, 21, 28, and 42, the mean PCS score was significantly (*p* < 0.05) reduced from baseline in both the TP and TP plus DLX groups. The reduction in the PCS score from baseline to day 7 (−3.71 vs. −3.7; *p* = 1.89), day 14 (−8.49 vs. −8.9; *p* = 1.01), day 21 (−17.74 vs. −15.86; *p* = 0.76), day 28 (−26.77 vs. −25.6; *p* = 0.89), and day 42 (−28.51 vs. −28.83; *p* = 1.11) was not significantly different in the TP group compared to the TP plus DLX group (Table 2).

#### 3.3.7. Adverse Events

The percentage of patients who experienced at least one AE was similar in the TP (84.6% [44/52]) and TP plus DLX (85.7% [48/56]) treatment groups. AEs reported by patients in either treatment group are summarized in Table 3. A total of 114 AEs were reported: 55 AEs were reported in TP group and 59 AEs in TP plus DLX group. No serious AEs occurred during the period of observation. The AEs incidence was not significantly different in the TP group compared to the TP plus DLX group. The AEs reported in TP and TP plus DLX groups were nausea (23.1 vs. 21.4%; *p* = 0.97), vomiting (9.6 vs. 0%; *p* < 0.05), dry mouth (7.7 vs. 16.1%; *p* < 0.05), fatigue (5.7 vs. 12.5%; *p* < 0.05), insomnia (13.5 vs. 10.7%; *p* = 0.92), dizziness (15.4 vs. 10.7%; *p* = 0.88), constipation (11.5 vs. 8.9%; *p* = 0.98), diarrhea (5.7 vs. 8.9%; *p* = 0.91), drowsiness (0 vs. 7.1%; *p* < 0.05), decreased appetite (5.7 vs. 5.3%; *p* = 1.21), hyperhidrosis (0 vs. 3.5%; *p* = 0.90), and pruritus (7.7 vs. 0%; *p* < 0.05). No association between side-effect incidence and dose increasing was observed.

## 4. Discussion

In this randomized, double-blind trial comparing TP and TP plus DLX for CIPN management, TP was feasible and non-inferior to the association with DLX as far as the reduction of pain after chemotherapy at 28 days is concerned.

The experience of pain depends on the central processing of ascending (incoming) signals from peripheral tissues, which are modulated by descending inhibitory and facilitatory mechanisms.

The origin of supraspinal (or descending) pain control pathways is from different supraspinal sites. Descending pain control pathways can be facilitatory or inhibitory. They influence the perception of pain and modulate the activity of spinal nociceptors regulating the propagation of signals to the brain. The release of neurotransmitters, such as endogenous opioids, norepinephrine, and serotonin, is involved in endogenous pain modulation. The μ-opioid receptor (MOR) agonists inhibit transmission of pain signals along the ascending pathways and are involved in the modulation of supraspinal pain signals through their action in descending pathways. Norepinephrine inhibits pain through α2 adrenoceptors, while serotonin facilitates or inhibits functions depending on the receptor subtype.

Tapentadol acts both as a MOR agonist and as a norepinephrine reuptake inhibitor (NRI), thereby generating synergistic analgesic action [9,10].

The clinical efficacy of the tapentadol has previously been established in patients with diverse types of painful peripheral neuropathies (PPN), such as postherpetic neuralgia and painful diabetic polyneuropathy [11]. Galiè et al. conducted an open-label, 3-month, prospective study assessing the efficacy of tapentadol in 31 patients with moderate-to-severe neuropathic pain from a CIPN. It was unresponsive to maximum doses of antineuropathic antidepressants and anticonvulsants [12].

According to ASCO guidelines, duloxetine is recommended for clinical practice in patients with painful CIPN [8].

Notably, the present trial showed that tapentadol-treated patients versus tapentadol plus duloxetine recipients with CIPN experienced similar NRS and DN4 pain scores after 28 and 42 days of treatment.

Scores on other rating scales—HADS, QLQ-CIPN20, PRS, and PCS—also revealed similar improvements associated with tapentadol versus duloxetine at these time points.

CINP negatively affects psychological distress and sleep quality in cancer patients treated with chemotherapy [28]. Considering the increasing emphasis given to the quality of life among cancer patients, the association between CIPN and QoL has been largely explored. CIPN negatively influences the QoL of cancer patients by producing unpleasant symptoms, such as anxiety, depression, and sleep disorders [28].

Our data suggest that CIPN has a negative effect on psychological distress and sleep quality in patients treated with chemotherapy. In this study, enhanced QoL is associated with functional recovery and improved sleep quality. This beneficial action of TP probably depends on its peculiar pharmacological action, which is characterized also by a marked noradrenergic activity.

AEs were mild and tolerable. Tapentadol shows fewer typical opioid-induced side effects compared with equianalgesic doses of classical opioids [29]. This favorable tolerability profile is associated with maintenance of QoL and improved compliance.

Neurophysiological endpoints were also assessed via LEPs for studying Aδ and C nerve-fiber transmission involved in afferent nociception [16,18,19,30,31,32]. Heat impulses produced by a laser were used to selectively stimulate nerve terminals in the epidermis, activate Aδ and C nerve fibers, and evoke potentials recordable via electrodes on the scalp.

LEPs data from our trial in patients with CINP treated with tapentadol or duloxetine revealed a significant reduction in latency and a significant increase in amplitude of N2 and P2 potentials.

We believe that the marked analgesic efficacy of tapentadol, together with available safety data, support the use of this drug for the treatment of CIPN.

### Limitations

Our study should minimize most of the common biases. Nevertheless, our study has a number of limitations. Despite the random assignment, there was imbalance between the study arms in the male/female ratio.

## 5. Conclusions

The objective of this randomized non-inferiority trial was to demonstrate the efficacy of the tapentadol in reducing the intensity of chemotherapy-induced peripheral neuropathy in patients treated with chemotherapeutic agents. Detailed analysis of the study findings revealed that the study objective was met; that is, the use of the TP is a safe and effective analgesic therapy in patients with CIPN. Clinical efficacy of this formulation was mediated by its action as a μ-opioid receptor agonist and as a norepinephrine reuptake inhibitor. In addition, positive effects of the TP were noted on the patient QoL. Overall, given the high incidence of chemotherapy-induced peripheral neuropathy and the few therapeutic tools available, the clinical value of a safe and effective analgesic therapy, such as the tapentadol, which also considerably improves the patient QoL, is clear.

## Figures and Tables

**Figure 1 cancers-14-04002-f001:**
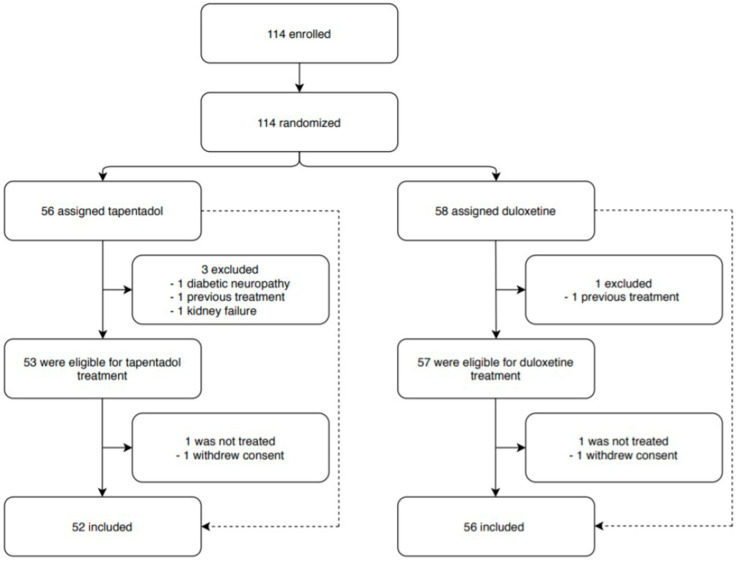
Trial profile.

**Figure 2 cancers-14-04002-f002:**
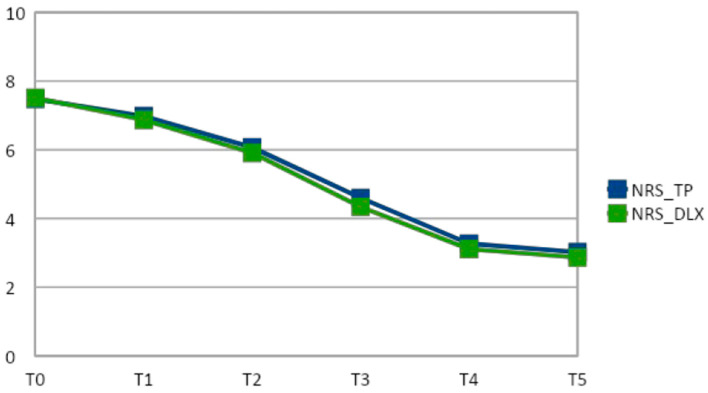
NRS in TP and TP plus DLX groups (x: NRS, y: time points).

**Figure 3 cancers-14-04002-f003:**
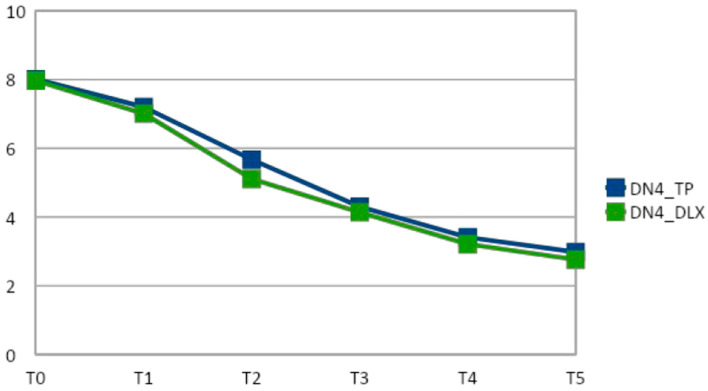
DN4 in TP and TP plus DLX groups (x: DN4, y: time points).

**Table 1 cancers-14-04002-t001:** Demographic and baseline characteristics of the 108 enrolled patients.

	TP (n = 52)	TP + DLX (n = 56)
**Age at enrollment** (years)	52.4 (34–82)	51.7 (29–81)
**Gender**, n (%)		
Male	20 (38.5)	22 (39.3)
Female	32 (61.5)	34 (60.7)
**Tumour type**, n (%)		
Non-solid	5 (9.6)	7 (12.5)
Solid	47 (90.4)	49 (87.5)
**Solid tumour**, n (%)		
Breast	19 (40.4)	18 (36.7)
Digestive system	13 (27.6)	14 (28.6)
Respiratory system	6 (12.8)	6 (12.2)
Gynecological apparatus	5 (10.6)	6 (12.2)
Oropharynx	4 (8.6)	5 (10.3)
**Surgery treatment**, n (%)		
Yes	46 (88.5)	49 (87.5)
No	6 (11.5)	7 (12.5)
**Radiotherapy treatment**, n (%)		
Yes	21 (40.4)	22 (39.3)
No	31 (49.6)	34 (50.7)
**Chemotherapy regimen**, n (%)		
Taxanes based	22 (42.3)	24 (42.8)
Platinum based	18 (34.6)	19 (33.9)
Taxanes + Platinum	5 (9.6)	5 (8.9)
Others	7 (13.5)	8 (14.4)

**Table 2 cancers-14-04002-t002:** Study data at baseline (T0) and at 7 (T1), 14 (T2), 21 (T3), 28 (T4), 42 (T5) days after treatment.

	TP (n = 52)						TP + DLX (n = 56)					
	T0	T1	T2	T3	T4	T5	T0	T1	T2	T3	T4	T5
**NRS**	7.48	6.98	6.07	4.60	3.27	3.02	7.51	6.87	5.91	4.34	3.11	2.87
**DN4**	8.01	7.20	5.67	4.30	3.41	2.98	7.97	7.01	5.11	4.14	3.21	2.76
**HADS**	15.34	13.23	11.06	9.23	6.74	6.20	15.11	12.96	10.57	9.12	6.81	6.42
**QoL**	69.46	63.28	52.09	44.58	33.15	28.76	70.13	62.31	50.12	41.19	31.89	26.58
**PRS**												
**Complete relief**	-	0	0	8 (15.4)	27 (51.9)	35 (67.3)	-	0	0	13 (23.2)	30 (53.6)	39 (69.6)
**Much relief**	-	9 (17.3)	16 (30.8)	34 (65.4)	20 (38.5)	16 (30.8)	-	10 (17.8)	21 (37.5)	30 (53.6)	18 (32.1)	16 (28.6)
**Mild relief**	-	41 (78.8)	36 (69.2)	10 (19.2)	5 (9.6)	1 (1.9)	-	44 (78.6)	35 (62.5)	13 (23.2)	8 (14.3)	1 (1.8)
**No change**	-	2 (3.8)	0	0	0	0	-	2 (3.6)	0	0	0	0
**LEPs**												
**N2 latency (ms)**	257.6	-	-	-	-	224.8	255.1	-	-	-	-	221.6
**N2 amplitude (μV)**	−28.7	-	-	-	-	−15.2	−27.5	-	-	-	-	−14.8
**P2 latency (ms)**	427.3	-	-	-	-	396.8	430.2	-	-	-	-	397.4
**P2 amplitude (μV)**	10.6	-	-	-	-	22.4	11.8	-	-	-	-	23.1
**PCS**	41.63	37.92	33.14	23.89	14.86	13.12	41.81	38.11	32.91	25.95	16.21	12.98

**Table 3 cancers-14-04002-t003:** Adverse Events (AEs) during treatment with tapentadol and duloxetine.

AEs	TP (n = 55)	TP + DLX (n = 59)	*p*
**Nausea**	12 (23.1)	12 (21.4)	0.97
**Vomiting**	5 (9.6)	–	<0.05
**Dry mouth**	4 (7.7)	9 (16.1)	<0.05
**Fatigue**	3 (5.7)	7 (12.5)	<0.05
**Insomnia**	7 (13.5)	6 (10.7)	0.92
**Dizziness**	8 (15.4)	6 (10.7)	0.88
**Constipation**	6 (11.5)	5 (8.9)	0.98
**Diarrhea**	3 (5.7)	5 (8.9)	0.91
**Drowsiness**	–	4 (7.1)	<0.05
**Decreased appetite**	3 (5.7)	3 (5.3)	1.21
**Hyperhidrosis**	–	2 (3.5)	0.90
**Pruritus**	4 (7.7)	–	<0.05
**TOTAL**	**55**	**59**	

## Data Availability

The datasets generated during the current study are available from the corresponding author on reasonable request.

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
