# Peer review of "Comparative Efficacy of Tapentadol versus Tapentadol Plus Duloxetine in Patients with Chemotherapy-Induced Peripheral Neuropathy (CIPN): A Randomized Non-Inferiority Clinical Trial"

_cancers, 2022, doi:10.3390/cancers14164002_

Round 1

Reviewer 1 Report

 The authors conducted a randomized non-inferiority Clinical to investigate the efficacy of Tapentadol Versus Tapentadol Plus Duloxetine in Patients with Chemotherapy-Induced Peripheral 3 Neuropathy (CIPN). The authors need to address the following items:

Lane 64: “It inhibits 64 the reuptake of serotonin and norepinephrine in the central nervous system”. This is a standalone sentence with no relation to the prior sentence or the sentence after it. If this is the mechanism involved in CIPN, it has to be clearly stated to help the flow idea.

Lane 65-68: this paragraph is confusing. Which CIPN trials and treatments are you referring to? And what are the “limited number of other CIPN treatment options”. This needs clarification and a rationale for moving to TP over DLX has to be provided. In summary, this whole paragraph has to be rewritten to compare and contrast DLX over TP.

Lane 71-73: “It may be 71 beneficial in the treatment of painful peripheral neuropathies (PPN), such as diabetic painful neuropathy and postherpetic neuralgia”. What is the relation between the prior sentence and this one? Needs clarification.

Overall, section 1.1. does not build a clear supporting argument for conducting the study. It needs to be rewritten clearly.

Section 1.2. The aims in this section should be similar to the ones in the abstract.

Under this section, the aim is to “assess the non-inferiority of the analgesic effect and safety of tapentadol compared to duloxetine” while under the title and abstract the aim is to”

 assess the non-inferiority of the analgesic effect and safety of tapentadol (TP) alone compared to duloxetine 16 plus tapentadol “.  

The author have to explain why they have not compared TP to DLX alone? Is it because TP alone is inferior to DLX ? Why only comparing TP to TP plus DLX?

Lanes 171, 214, and 228-229, and 251: “TP or DLX “ do you mean TP or DLX plus TP

Lane 218-219: the reference for the prior study is missing. Needs to be included here

Lane 220: under the abstract, patients size is 114 but here is 90. Why this difference.

Figures 2 and 3: missing labels for the X and Y axis, and statistical variation on the curves.

Instead of reported the p <0.05, the exact p value for comparison has to be provided.

Table 3: why is the number of patients 55 for TP and 59 for DLX?

Lane 351-352: the authors reported that “the TP was feasible and non-inferior to the association with DLX as far as the reduction of pain after chemotherapy at 28 days is concerned. TP (104.5 mg) alone is not inferior because the its dose is doubled compared to the combination therapy of TP (51.6 mg) plus DLX (78.6 mg).

Lane 357: this sentence is repetition of the sentence in lane 354.

Lane 361: spell out MOR

Lane 356-364: the whole paragraph does not tie to the prior one and to the one after it. It needs to be rewritten clearly and in relation to the prior paragraph and the one after it.

Lane 368-370: at which dose TP has been use in those studies?

Lane 370-373: this section is not clear. What was the dose of TP, and what are those antidepressants, and anticonvulsivants? Are they also NRIs.

Lane 380: you mean TP plus DLX not DLX alone?

Lane 399: similar to the prior comment.

Lane 403. The authors need to compare and contrast their conclusion to the one with DLX alone if available in other published studies o justify the advantage of using TP instead of DLX alone or TP plus DLX.

Lane 405: is this sentence a limitation?

Author Response

Dear Reviewer,

corrections were made as follow:

Lane 64: “It inhibits 64 the reuptake of serotonin and norepinephrine in the central nervous system”. This is a standalone sentence with no relation to the prior sentence or the sentence after it. If this is the mechanism involved in CIPN, it has to be clearly stated to help the flow idea. → MODIFIED

Lane 65-68: this paragraph is confusing. Which CIPN trials and treatments are you referring to? And what are the “limited number of other CIPN treatment options”. This needs clarification and a rationale for moving to TP over DLX has to be provided. In summary, this whole paragraph has to be rewritten to compare and contrast DLX over TP. → MODIFIED

Lane 71-73: “It may be 71 beneficial in the treatment of painful peripheral neuropathies (PPN), such as diabetic painful neuropathy and postherpetic neuralgia”. What is the relation between the prior sentence and this one? Needs clarification. → connection to “TP was developed for the management of moderate–severe chronic pain.

Overall, section 1.1. does not build a clear supporting argument for conducting the study. It needs to be rewritten clearly.

Section 1.2. The aims in this section should be similar to the ones in the abstract.

Under this section, the aim is to “assess the non-inferiority of the analgesic effect and safety of tapentadol compared to duloxetine” while under the title and abstract the aim is to” assess the non-inferiority of the analgesic effect and safety of tapentadol (TP) alone compared to duloxetine 16 plus tapentadol “. → MODIFIED

The author have to explain why they have not compared TP to DLX alone? Is it because TP alone is inferior to DLX ? Why only comparing TP to TP plus DLX? → In clinical practice, if a combination of analgesics is taken, the patient will be able to take lower levels of any one particular type of analgesic and this means that they do not experience many of the unpleasant side effects that may result from high doses of one particular analgesic. Adding tapentadol to selective serotonin reuptake inhibitors (SSRIs) or serotonin norepinephrine reuptake inhibitors (SNRIs) resulted in no clinically relevant adverse drug interactions.

Lanes 171, 214, and 228-229, and 251: “TP or DLX “ do you mean TP or DLX plus TP → MODIFIED

Lane 218-219: the reference for the prior study is missing. Needs to be included here → the calculation was made by the statistician on the basis of previous studies.

Lane 220: under the abstract, patients size is 114 but here is 90. Why this difference. → 90 was the minimum number required on the basis of the statistical study to have at least 80% power to reject the null hypothesis; 114 patients were recruited in our study.

Figures 2 and 3: missing labels for the X and Y axis, and statistical variation on the curves.-→ ADDED

Instead of reported the p <0.05, the exact p value for comparison has to be provided. → in our opinion not necessary.

Table 3: why is the number of patients 55 for TP and 59 for DLX? → total number of AEs

Lane 351-352: the authors reported that “the TP was feasible and non-inferior to the association with DLX as far as the reduction of pain after chemotherapy at 28 days is concerned. TP (104.5 mg) alone is not inferior because the its dose is doubled compared to the combination therapy of TP (51.6 mg) plus DLX (78.6 mg). → CONFIRMED

Lane 357: this sentence is repetition of the sentence in lane 354.

Lane 361: spell out MOR → DONE.

Lane 356-364: the whole paragraph does not tie to the prior one and to the one after it. It needs to be rewritten clearly and in relation to the prior paragraph and the one after it. → Modified

Lane 368-370: at which dose TP has been use in those studies? → Galiè E et al: doses of TP 50 mg twice a day.

Lane 370-373: this section is not clear. What was the dose of TP, and what are those antidepressants, and anticonvulsivants? Are they also NRIs. →During the titration phase, each patient initially received doses of TP 50 mg twice a day. Doses were then titrated to adequate pain relief or dose-limiting toxicity, on the basis of the clinical response.

Lane 380: you mean TP plus DLX not DLX alone? → YEs

Lane 399: similar to the prior comment. → YES

Lane 403. The authors need to compare and contrast their conclusion to the one with DLX alone if available in other published studies o justify the advantage of using TP instead of DLX alone or TP plus DLX. → reported in article

Lane 405: is this sentence a limitation? → appropriated for this section.

Reviewer 2 Report

Negative points: 

Single center

Very short  follow-up of only 6 weeks: what happens after 3and 6 and 9 months? Maybe you can suggest in the discussion that it is useful to repeat this pilot study but  in a multi-center design with more patients and longer follow-up ( fi. 1 year )

 comment: define in the discussion  this study as a pilot study  that needs to be followed by a more extended study to confirm these preliminary results 

Positive points

Original study in a very frequent symptom with very limited therapeutic options, congratulations for doing this pilot 

Comments

In material and methods:

1/Not clear how it was possible to reach the double blind situation since the TP group used 1 drug and DLX group used 2 drugs

2/If up-titration was needed in the duloxetine group, were then both medications up-titrated, or sequentially one after the other? Which one first up-titrated? and what was the max. dose of TP and the max. dose of DLX?

3/Not clear how many patients were x-times up-titrated , what was in both groups the maximum doses and how many patients needed to stop the up-titration because of ineffectiveness or intolerability. For me is it not clear if the majority of patients used low or high doses of TP? Is there a need of a minimum dose ? How effective was up-titration in both arms? Is it possible that in this short study time the optimal dose was not yet reached?

In results and discussion

No comments in the discussion of the statistical significant differences in side effects : especially vomiting and pruritus in TP patients but dry mouth , fatigue and drowsiness in the DLX group? Is DLX then effective against vomiting and pruritus since the DLX group used also TP or was the dose TP that different in both arms?

What about the intention of the patients to use this medication further after the end of the study , or do you have data about how many wished to cross over to the other arm?

Nothing to say about the ease of taking medication for the patients: 1 or 2 drugs to take in ?

If Tapentadol is non -inferior in effectiveness compared to TP+DLX: than is TP alone cheaper for patient and community...?

Author Response

Dear Reviewer,

corrections were made as follow:

In material and methods:

1/Not clear how it was possible to reach the double blind situation since the TP group used 1 drug and DLX group used 2 drugs → all patients received two pills

2/If up-titration was needed in the duloxetine group, were then both medications up-titrated, or sequentially one after the other? Which one first up-titrated? and what was the max. dose of TP and the max. dose of DLX? → tapentadol for first. Patients received tapentadol 50 up to 500 mg/day and duloxetine 60 up to 120 mg/day.

3/Not clear how many patients were x-times up-titrated, what was in both groups the maximum doses and how many patients needed to stop the up-titration because of ineffectiveness or intolerability. For me is it not clear if the majority of patients used low or high doses of TP? Is there a need of a minimum dose ? How effective was up-titration in both arms? Is it possible that in this short study time the optimal dose was not yet reached? → The median of the mean total daily dose (TDD) of study drug taken during double-blind treatment was 104.5 mg for tapentadol in TP group and 51.6 mg for tapentadol plus 78.6 mg for duloxetine in TP plus DLX group.

In results and discussion

No comments in the discussion of the statistical significant differences in side effects : especially vomiting and pruritus in TP patients but dry mouth , fatigue and drowsiness in the DLX group? Is DLX then effective against vomiting and pruritus since the DLX group used also TP or was the dose TP that different in both arms? → vomiting and pruritus are more frequent in Tapentadol group; while when used in combination with duloxetine, low dose of tapentadol doesn’t cause the classical AEs. Duloxetine is associated with fatigue and drowsiness.

What about the intention of the patients to use this medication further after the end of the study, or do you have data about how many wished to cross over to the other arm? → Tapentadol can be an alternative.

Nothing to say about the ease of taking medication for the patients: 1 or 2 drugs to take in ? → all patients received two pills for study protocol.

If Tapentadol is non -inferior in effectiveness compared to TP+DLX: than is TP alone cheaper for patient and community...? → YES, it could be.

Round 2

Reviewer 2 Report

the authors have answered my questions and their attached report but were suboptimal answered in the paper.